# Reliability of the Linear Measurement (Contact) Method Compared with Stereophotogrammetry (Optical Scanning) for the Evaluation of Edema after Surgically Assisted Rapid Maxillary Expansion

**DOI:** 10.3390/healthcare8010052

**Published:** 2020-03-01

**Authors:** Gülperi Koçer, Samed Sönmez, Yavuz Findik, Tayfun Yazici

**Affiliations:** 1Oral and Maxillofacial Surgeon, Süleyman Demirel University, 32600 Isparta, Turkey; yavuzfindik32@hotmail.com (Y.F.); tayfunyazici@yahoo.com (T.Y.); 2Oral and Maxillofacial Surgeon, Private Practice, 07100 Antalya, Turkey; samedsonmez@hotmail.com

**Keywords:** facial swelling, linear contact method, stereophotogrammetry, SARME

## Abstract

Many techniques have been developed to evaluate facial swelling after maxillofacial surgeries. Patients who undergo surgically assisted rapid maxillary expansion (SARME) develop facial edema more often than those who undergo minor oral surgeries. Reliable systems to assess soft tissue dimensions offer many advantages for documentation and treatment planning across surgical fields. (1) Background: The objective assessment of facial swelling is advantageous as it allows the evaluation of the effect of anti-inflammatory drugs. Therefore, this study aimed to compare the reliabilities of linear measurement method and optical scanning for the objective assessment of facial swelling after SARME. (2) Methods: Sixteen (12 women and 4 men) patients were enrolled. Linear measurements between guide points and facial scans were obtained for the left and right sides preoperatively and 1, 2, and 5 days after SARME. Preoperative values were subtracted from each post-operative value and the differences were compared between the two measurement methods. (3) Results: There were no statistically significant differences between the right and left sides at any time point in the measurements with either method. (4) Conclusions: Recently, stereophotogrammetry has been considered the first choice method for evaluating facial swelling. Furthermore, we found a strong correlation between volumetric analysis and linear measurement at all time points and for both sides.

## 1. Introduction

Changes in the facial contour may result from craniofacial or orthognathic surgeries, inflammation, trauma, or ablative surgery. Evaluation of post-operative edema in the early period is important in terms of giving an idea about inflammation. In this case, the number of drugs that may have side effects such as NSAIA and methylprednisolone or dexamethasone, which should be used to remove or reduce inflammation-related swelling post-operatively, can be reduced. Several methods, mainly including contact measurement methods, have been used in the past 60 years to measure facial deformities of various types. [1,2]

The linear measurement (contact) method is inexpensive, simple, non-invasive, and can be performed with flexible rulers. It has a high applicability and can detect edema in tissues with high accuracy. However, the non-contact measurement method has been increasingly replacing it, although the newer methods often require complicated equipment for measurement to allow for the standard orientation of the head during photography and radiography [2,3]. Mathematical methods have also been applied to describe changes in the facial morphology [2,4]. Early (non-digital) stereophotogrammetry has been used to obtain linear measurements based on landmark positions [2,5,6].

Three-dimensional (3D) scanners are the first choice in studies on measurements and comparisons of volume. The most commonly used optical 3D scanners are the laser scanner, structured light scanner, and stereophotographic scanner [7]. Stereophotogrammetry is a technique in which 3D images are obtained using two or more cameras. The facial morphology can be generated with small margins of error on stereophotogrammetry. The symmetry, length, surface area, and volumetric differences can be calculated with digital overlapping of the created 3D objects [8,9].

Surgically assisted rapid maxillary expansion (SARME) is a technique used to correct transverse skeletal discrepancies. When applied for appropriate indications, it provides predictable and successful results [10].

Haas reported that after cessation of growth and complete ossification of the midpalatal suture, conventional RME has limited efficacy, as the maxilla resists expansion [11]. SARME was developed to eliminate this disadvantage. It mainly produces controlled soft tissue expansion with distraction osteogenesis and surgical procedures, including osteotomy of the pyriform aperture to the pterygomaxillary suture and midpalatal osteotomy [12]. 

The aim of the present study was to evaluate and compare reliabilities of the linear measurement method and optical scanning for the objective assessment of facial swelling after SARME.

## 2. Materials and Methods 

Since long cut lines are made in bone and soft tissue in SARME, post-operative edema is expected to be high. In the power analysis conducted by taking into consideration the previous literature studies, it was determined that the number of experiment units that should be in each group should be 16 with 95% power. Therefore, the study was planned in sixteen patients (4 men and 12 women) aged 18–24 years scheduled to undergo SARME were enrolled in the study. The study was approved by the Süleyman Demirel University Research Ethics Board (21.01.2020, 72867572-050-11449) and was supported by the TUBITAK (Turkey Science and Technic Investigation Association) 3001 support program (project number: 115-S-153). All experiments on human subjects were conducted in accordance with the Declaration of Helsinki (http://www.wma.net) and all patients agreed to participate in the study and provided written informed consent before undergoing the treatment and fulfilled the eligibility criteria. Patients with systemic diseases, congenital craniofacial abnormalities, pregnancy, beard or mustache hindering 3D analyses, age below 18 years, and malignancies were excluded.

All surgeries were performed under general anesthesia by the same surgical team. Bilateral osteotomies were performed from the piriform rims to the pterygomaxillary junctions with a piezoelectric device. The pterygomaxillary junctions were not released. A sagittal osteotomy was performed, running from the midline of the alveolar bone to the junction between the central incisors. During the surgery, all patients received 1 mg/kg of methylprednisolone, 1 g of penicillin, and 75 mg of diclofenac sodium. Furthermore, all patients received antibiotics (1 g amoxicillin and clavulanate potassium) every 12 h after the surgery for 5 days. Paracetamol was prescribed to all patients as an analgesic to be taken when needed. Anti-inflammatory drugs were not prescribed because they could alter the volume of the edema to different extents among patients who took different drug dosages. 

In the preoperative period, distances between various groups of guide points in all patients were measured with a flexible tape. All evaluations were performed by a single physician. The patients were placed in the upright position and instructed to close the mouth. Resting measurements were obtained with no gestures, preoperatively (T0) and 1 (T1), 2 (T2), and 5 days (T3) post-operatively. The distance between the bilateral tragus rim corners and between the outer canthus of the eye and the angle of the mandible, defined by Amin and Laskin, were modified. The outer canthus, corner of the mouth, anterior edge of the tragus, and angle of the mandible were selected as guide points [13].

Distances between the anterior edge of the tragus and the corner of the mouth (D1), between the corner of the mouth and the angle of the mandible (D2) (Appendix A), between the outer canthus of the eye and the angle of the mandible (D3), and between the outer canthus of the eye and the corner of the mouth (D4) (Appendix A) were measured in cm and recorded. The distances were measured separately for the right and left sides. 

The total right–left facial length was calculated as the sum of the four corresponding values on the right and left sides. The mean facial length was the total right–left facial length divided by 4. The total facial length was the sum of the mean facial lengths.

Mean facial length = (D1 + D2 + D3 + D4)/4

Total facial length = Right mean facial length + Left mean facial length

The obtained values were recorded, and the measured values at T0 were subtracted from those at T1, T2, and T3.

The 3dMD (DSP400® stereo-optical 3-dimension) optical scanner was used to obtain 3D facial scans. It can capture full facial images from one ear to another and including the chin in 2 ms at the highest resolution. The geometrical accuracy of the facial system used in the study was < 0.2 mm, as claimed by the manufacturers. To evaluate facial swelling, facial scans were uploaded in the 3dMD Vultus software (3dMD Vultus® software, version 2.2.0.18, 3dMD, Atlanta, GA 30339, USA); scans were obtained preoperatively and 1, 2, and 5 days post-operatively before performing distraction osteogenesis. 

In the whole-face screening process, men were evaluated without beard and shoulders because these would have affected the evaluation of extraoral edema and hindered the 3D surface overlap. Women were evaluated without make-up, earrings, and speeding jewelry. Patients were placed perpendicular to the fixed point at a distance of 1.5 m from the 3D scanner. The head was positioned at an angle of 45° when the side profile was visualized for a full scan of the jaw and neck region. During screening, patients were instructed not to perform gestation movements at rest and to close their lips. Images containing polygon losses or a damaged surface were evaluated after shooting and were repeated, if necessary. Data were saved in the “stl.” vs “obj.” format. Subsequently, five reference points were identified, and changes in the volume among them were calculated on the left and right sides. The reference points were as follows: tragus, angle of the mandible; lateral canthus of the eye; corner of the mouth; and lateral side of the ala nasi. 

Measurements were obtained preoperatively and on post-operative days 1, 2, and 5 for each side. Differences between the preoperative values and the values obtained on post-poperative days 1, 2, and 5 were calculated. 

All 3D calculations were performed thrice by the same person. In cases of a difference between the same measurement at different attempts of more than 10%, the measurements were repeated. The arithmetic mean of the three measurements was calculated and recorded. Differences in the volume between commissures and lateral sides of ala nasi were estimated. All volumetric differences were recorded in dL.

Statistical analyses were performed using the Statistical Package for the Social Sciences software, version 15.0 (2006). After checking the data obtained for the characteristics of the parametric tests, the appropriate homogeneity of variance was confirmed with the Box-M test and normal distribution was confirmed with the Anderson–Darling test. Measurements were repeated for the time factor. A p-value < 0.05 was considered statistically significant. For the comparison between the measurement methods, the intergroup correlation was examined. 

## 3. Results

There were no statistically significant differences between the right and left sides at any time point in the measurements with either method (*p* > 0.05) (Table 1). 

The total volume of the edema was measured, and the highest value was observed on post-operative day 2 (*p* < 0.05). The volume of edema on day 5 was less than that on day 1, but there was no statistically significant difference (*p* < 0.05). The volume of edema on post-operative day 2 was significantly higher than that on post-operative days 1 and 5 (*p* < 0.05) (Table 1.).

The volumes of edema on the right and left sides were similar on post-operative days 1, 2, and 5 based on the values obtained using the guide points. The volume of edema was the highest on both sides on post-operative day 2 and lowest on post-operative day 5. While the volume of edema showed no statistically significant difference between post-operative days 1 and 5, it was significantly higher in both groups on post-operative day 2 (*p* < 0.05) (Table 1.).

There were no statistically significant differences between the right and left sides at any time point in terms of the mean of all volumetric measurements with either method (*p* > 0.05) (Table 2.).

In the volumetric measurements of edema, the highest volume was on post-operative day 2 (*p* < 0.05). The volume of edema on post-operative day 5 was less than that on post-operative day 1, but there was no statistically significant difference (*p* > 0.05). The volume of edema on post-operative day 2 was significantly higher than that on post-operative days 1 and 5 (*p* < 0.05) (Table 2.).

The volumes of edema on the right and left sides were symmetrical on post-operative days 1, 2, and 5, irrespective of the group (*p* > 0.05). The volume of edema was the highest on both sides on post-operative day 2 and the least on post-operative day 5 (Table 2.).

Correlation coefficients were evaluated for volumetric values and measurements obtained using guide points, and a strong correlation was found between the two measurement methods at T1, T2, and T3 (Table 3.).

## 4. Discussion

The measurement of distances between various groups of guide points on the face is the most widely used technique in terms of applicability, repeatability, and ease. This method was proposed by Amin and Laskin to measure the volume of edema after mandibular tooth extraction; we modified it considering that the edema may intensify in the middle region of the face. Recently, volumetric evaluation on stereophotogrammetry has been frequently performed in the fields of medicine and dentistry [14,15,16] In addition, stereophotogrammetry has been used in many studies, such as those on edema assessment after a third molar extraction and on long-term evaluation of facial volumetric differences after orthognathic surgeries [17,18,19,20].

The volume of facial edema measured on stereophotogrammetry is considered to be the closest to the actual volume, obtained in numerical values with lesser margins of error. Van der Meer reported that stereophotogrammetry was reliable and valid for the measurement of edema greater than 5.9 dL in volume [7]. Yip placed 7.5- and 10-dL artificial apparatus on the face and examined the volumetric changes with the stereophotometric method and reported that the margin of error was approximately 1% [21]. None of the 288 volumetric measurement differences in our study were below 5.9 dL, and all were above the lower confidence limit set by van der Meer [7].

With facial scans obtained using lasers or LED, a 3D analysis of the face can be performed using software programs, and volumetric differences can be calculated. Although it is an effective method, the relatively long duration of scanning can cause deformities due to small movements. Another disadvantage is the cost associated with the device. The linear measurement (contact) method is the oldest and most commonly used technique, which meets the minimum requirements of clinical studies on objective assessments of post-operative facial swelling or other induced changes in facial dimensions. 

The aim of this study was to investigate the correlation between the measurement of distances between different groups of guide points and to detect facial edema on stereophotogrammetry, which has been considered to provide the most accurate data in the volumetric measurement of edema. Based on the results of our study, a strong correlation was observed between the two measurement techniques at all time points (98–99%). Therefore, it can be concluded that measurements performed using guide points on the face are effective, less time-consuming, cost-effective, and highly feasible in evaluating facial edema. However, the correlation between the measurement methods was only evaluated for the new measurement technique that we have developed; hence, we believe that there is a need to compare different modified measurement techniques with stereophotogrammetry.

## 5. Conclusions

Recently, stereophotogrammetry has been considered as the first choice modality for evaluating facial swelling. This study revealed a strong correlation between the volumetric analysis and linear measurements. Therefore, linear measurement is an easy, reproducible, inexpensive method to measure the volume of facial edema.

## Figures and Tables

**Table 1 healthcare-08-00052-t001:** Difference in the mean facial length between the preoperative and post-operative values at different time points.

Side	Day 1 (cm)	Day 2 (cm)	Day 5 (cm)	Mean (cm)
Right	0.64 ± 0.02	0.72 ± 0.03	0.59 ± 0.023	0.65 ± 0.02
Left	0.63 ± 0.028	0.71 ± 0.032	0.59 ± 0.026	0.64 ± 0.02
Both	1.26 ± 0.02	1.42 ± 0.03	1.42 ± 0.03	

**Table 2 healthcare-08-00052-t002:** Volumetric measurements of edema at different time points.

Side	Day 1 (dL)	Day 2 (dL)	Day 5 (dL)	Mean (dL)
Right	12.634 ± 0.618	15.027 ± 0.922	10.994 ± 0.648	12.885 ± 0.616
Left	12.183 ± 0.839	15.027 ± 0.922	10.859 ± 0.779	12.469 ± 0.801
Both	24.818 ± 0.684	29.392 ± 0.926	21.852 ± 0.7	

**Table 3 healthcare-08-00052-t003:** Correlation coefficients.

		Face Volumetric		
		T1	T2	T3
**General Face Guide Points**	**1**	0.99		
	**2**		0.99	
	**3**			0.98

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
