# Peer review of "Reliability of the Linear Measurement (Contact) Method Compared with Stereophotogrammetry (Optical Scanning) for the Evaluation of Edema after Surgically Assisted Rapid Maxillary Expansion"

_healthcare, 2020, doi:10.3390/healthcare8010052_

Round 1

Reviewer 1 Report

"Reliability Of The Linear Measurement (Contact) 2 Method Compared With Spectrophotogrammetry 3 (Optical Scanning) For The Evaluation Of Edema After Surgically Assisted Rapid Maxillary Expansion.

The authors investigated the correlation between the 3D measurement and linear measurement of facial swelling after SARME in 16 patients. The result is clear but the purpose of this study seems unclear. Is the linear measurement performed in the present study novel? What is new should clearly be mentioned somewhere.

Furthermore it should be mentioned why SARME was selected to evaluate in this study. And it is not clear why the swelling must be evaluated during early post-surgery phase accurately.

The 16 patients were included for this study. The sample number may be not enough to discuss the correlation between 3D measurement and linear analysis. The power analysis should be performed before study properly.

In abstract, "Results: There were no statistically significant differences between the right and left sides at any time point in the measurements with either method." This is far from the purpose of the study, while the conclusion matches the purpose.

The figures to show how to measure should be presented visually, though the table showed the result data.

Author Response

Response to Reviewer 1 Comments

Dear Reviwer,

Thank you for your kind comments about the manuscript titled that "Reliability Of The Linear Measurement (Contact)  Method Compared With Spectrophotogrammetry  (Optical Scanning) For The Evaluation Of Edema After Surgically Assisted Rapid Maxillary Expansion. Your comments and suggestions will increase the value of our article.

Point 1: The authors investigated the correlation between the 3D measurement and linear measurement of facial swelling after SARME in 16 patients. The result is clear but the purpose of this study seems unclear. Is the linear measurement performed in the present study novel? What is new should clearly be mentioned somewhere.

Response 1: In the study I have done before, (G, Yuce E, Tuzuner Oncul A, Dereci O, Koskan O. Effect of the route of administration of methylprednisolone on oedema and trismus in impacted lower third molar surgery. Int J Oral Maxillofac Surg. 2014 May;43(5):639-43. doi: 10.1016/j.ijom.2013.11.005. Epub 2013 Dec 12. PubMed PMID: 24332587) one of the referees criticized why I used the linear measurement method to evaluate the postoperative edema and why I did not measure with 3D spectrophotogrammetry and so this study is planned when it is said that  the digital method gave more reliable results. Since the digital measuring device was expensive, it was not in our faculty at the time. Afterwards, when this device was taken to our faculty, I really wanted to evaluate whether the results of our previous  study are safe. In many institutions, digital measuring devices are not available because they are expensive. This study has been planned in order to reveal that in measurement of postoperative edema,  linear measurement method is still a reliable method compared to digital method, which is considered to be more objective.  The objective of the study explained in lines 63, 64  but if it is not enough It can be highlighted.

Point 2: Furthermore it should be mentioned why SARME was selected to evaluate in this study. And it is not clear why the swelling must be evaluated during early post-surgery phase accurately.

Response 2: It is mentioned that why SARME was selected to evaluate in this study in the 69th and 70th lines. It is corrected as your comments. ‘Evaluation of postoperative edema in the early period is important in terms of giving an idea about inflammation. In this case, the amount of drugs that may have side effects such as NSAIA and Methylprednisolone or dexamethasone, which should be used to remove or reduce inflammation-related swelling postoperatively can be reduced’ added to lines 38-41.

Point 3: The 16 patients were included for this study. The sample number may be not enough to discuss the correlation between 3D measurement and linear analysis. The power analysis should be performed before study properly.

Response 3:  In the power analysis conducted by taking into consideration the previous literature studies, it was determined that the number of experiment units that should be in each group should be 16 with 95% power. This mentioned in lines 70-72. Power analysis performed before the beginning of the study by the Assoc. Prof. Ozgur KOSKAN in  biometry  department who I mentioned in acknowledgement section.

Point 4: In abstract, "Results: There were no statistically significant differences between the right and left sides at any time point in the measurements with either method." This is far from the purpose of the study, while the conclusion matches the purpose.

Response 4: In this study both sides measurements  were compared with two measurement methods.  Only the results were mentioned in this part and the final  reports of the study were mentioned  in discussion and conclusion. It can be re-arranged as your suggestion.

Point 5: The figures to show how to measure should be presented visually, though the table showed the result data.

Response 5: In order to avoid giving repetitive information in accordance with the writing rules of the journal, the linear method measurement points are indicated in the table, the volume results in the figure.

Thank you for your concern.

Reviewer 2 Report

The paper is interesting but I have some questions.

Materials and methods

  • What were the inclusion criteria?
  • Why did the authors select Box-M test and Anderson-Darling test?
  • What correlation coefficient did the authors use?

Author Response

Response to Reviewer 2 Comments

Dear Reviwer,

Thank you for your kind comments about the manuscript titled that "Reliability Of The Linear Measurement (Contact)  Method Compared With Spectrophotogrammetry  (Optical Scanning) For The Evaluation Of Edema After Surgically Assisted Rapid Maxillary Expansion. Your comments and suggestions will increase the value of our article.

Point 1: What were the inclusion criteria?

Response 1: Inclusion criterias were to be over 18 and  to need SARME. Exclusion criterias were systemic diseases, craniofacial syndromes- congenital deformity, pregnancy, beard or mustache that make difficult to make 3D analysis, under the age of 18 and malignancy.

Point 2: Why did the authors select Box-M test and Anderson-Darling test?

Response 2: Box-M test was performed to control of the homogenity of variance and co-variance matrisis. Anderson-Darling was performed to control normal distribution. Although Shapiro-Wilk or Kalmogorov-Smirnov could be used Anderson-Darling was performed because of its more common   usage. 

Point 3: What correlation coefficient did the authors use?

Response 3: Repeated measurements were made on both time and direction factors. Pearson co-efficient is approximately 1. It can be added to the text as your suggestion if needed.

Thank you for your concern.

Round 2

Reviewer 1 Report

I think the authors have addressed my questions fully. The parameters the authors presented maybe useful for a certain situation. Congratulations on your work.